# Emergency Lung Transplantation after COVID-19: Immunopathological Insights on Two Affected Patients

**DOI:** 10.3390/cells10030611

**Published:** 2021-03-10

**Authors:** Giorgio A. Croci, Valentina Vaira, Daria Trabattoni, Mara Biasin, Luca Valenti, Guido Baselli, Massimo Barberis, Elena Guerini Rocco, Giuliana Gregato, Mara Scandroglio, Evgeny Fominskiy, Alessandro Palleschi, Lorenzo Rosso, Mario Nosotti, Mario Clerici, Stefano Ferrero

**Affiliations:** 1Department of Pathophysiology and Transplantation, University of Milan, 20122 Milan, Italy; valentina.vaira@unimi.it (V.V.); luca.valenti@unimi.it (L.V.); alessandro.palleschi@unimi.it (A.P.); lorenzo.rosso@unimi.it (L.R.); mario.nosotti@unimi.it (M.N.); mario.clerici@unimi.it (M.C.); 2Division of Pathology, Fondazione IRCCS Ca’ Granda, Ospedale Maggiore Policlinico, 20122 Milan, Italy; stefano.ferrero@unimi.it; 3Department of Biomedical and Clinical Sciences L. Sacco, University of Milan, 20157 Milan, Italy; daria.trabattoni@unimi.it (D.T.); mara.biasin@unimi.it (M.B.); 4Department of Transfusion Medicine and Hematology, Fondazione IRCCS Ca’ Granda Ospedale Maggiore Policlinico, 20122 Milan, Italy; guido.baselli@policlinico.mi.it; 5Department of Pathology, European Institute of Oncology, IRCCS, 20141 Milan, Italy; massimo.barberis@ieo.it (M.B.); Elena.Guerini@unimi.it (E.G.R.); 6Department of Oncology and Hemato-oncology, University of Milan, 20141 Milan, Italy; 7Division of Clinical Haematology-Oncology, European Institute of Oncology, IRCCS, 20141 Milan, Italy; giuliana.gregato@ieo.it; 8Department of Anesthesia and Intensive Care, San Raffaele Scientific Institute, Vita-Salute San Raffaele University, 20132 Milan, Italy; scandroglio.mara@hsr.it (M.S.); fominskiy.evgeny@hsr.it (E.F.); 9Unit of Thoracic Surgery and Lung Transplantation, Fondazione IRCCS Ca’ Granda Ospedale Maggiore Policlinico, 20122 Milan, Italy; 10Department of Molecular Medicine and Imaging in Rehabilitation, Don C. Gnocchi Foundation ONLUS, IRCCS, 20148 Milan, Italy; 11Department of Biomedical, Surgical and Dental Sciences, University of Milan, 20122 Milan, Italy

**Keywords:** SARS-CoV-2, COVID-19, ARDS, usual interstitial pneumonia, immunology, lung transplantation

## Abstract

We herein characterize the immunopathological features of two Italian COVID-19 patients who underwent bilateral lung transplantation (bLTx). Removed lungs underwent histopathological evaluation. Gene expression profiling (GEP) for immune-related signatures was performed on lung specimens and SARS-CoV-2-stimulated peripheral blood mononuclear cells (PBMCs). Cytokine levels were measured on lungs, bronchoalveolar lavage fluids and in culture supernatants. Pathological assessment showed extensive lung damage with the pattern of proliferative to fibrotic phases, with diffuse alveolar damage mimicking usual interstitial pneumonia (UIP). Lungs’ GEP revealed overexpression of pathogen recognition receptors, effector cytokines and chemokines, immune activation receptors and of the inflammasome components. Multiplex cytokine analysis confirmed a proinflammatory state, with high levels of monocyte/macrophage chemotactic and activating factors and of IL-6 and TNF-α. A similar profile was observed in SARS-CoV-2-stimulated PBMCs collected 7 days after transplant. The pattern of tissue damage observed in the lungs suggests that this may represent the output of protracted disease, resembling a diffuse UIP-like picture. The molecular immune profiling supports the paradigm of a persistent proinflammatory state and sustained humoral immunity, conditions that are maintained despite the iatrogenic immunosuppression.

## 1. Introduction

The clinical phenotype of coronavirus disease 2019 (COVID-19), following severe acute respiratory syndrome coronavirus 2 (SARS-CoV-2) infection, encompasses a wide spectrum of clinical manifestations, ranging from asymptomatic to severely complicated cases, with the highest rate of mortality due to acute respiratory distress syndrome (ARDS) [1,2,3,4]. At the pathological level, several elementary lesions can be documented in affected lungs, ranging from exudative diffuse alveolar damage (DAD) in the early phases to end-stage DAD and interstitial fibrosis, with the frequent presence of capillarostasis and microthrombi [5,6,7,8,9,10].

Current literature mostly focuses on acute or subacute stages, whereas only scant information is available on severely affected patients, where major tissue alterations likely lead to alterations of the lung function, possibly hampering the organ recovery [11].

The knowledge of the physiopathology of COVID-19 is still in its early phases, but a key mechanism seems to stem from the interaction between SARS-CoV-2 and angiotensin-converting enzyme 2 (ACE2), which is notably expressed by the epithelial layers of the upper airways and by the endothelial cells [12,13,14]. This interaction likely explains the elective tropism of SARS-CoV-2 for the respiratory district [9] and, possibly, the pathological features of neoangiogenesis and microangiopathy, with consequent complement activation and elicitation of an inflammatory cascade [15].

COVID-19 clinical evolution is highly variable [2,3,4], often with variegated multisystemic manifestations, but according to the current knowledge, the course of the disease is more predictable in elderly male patients, who require hospitalization at a higher frequency and suffer the highest rate of mortality. Severe cases affecting individuals younger than 40 years old constitute an almost negligible fraction of the affected population. Nevertheless, these younger patients represent a cohort where extreme lifesaving interventions such as lung transplantation could be envisioned [16].

Herein, we describe the histopathological and immunological features of lungs removed from the first two Italian patients, who successfully underwent emergency bilateral lung transplantations (bLTx) for COVID-19-related ARDS.

## 2. Materials and Methods

### 2.1. Patients

Two patients with extremely severe infection, whose lung functions were irreparably compromised, were selected to undergo bilateral lung transplantation at Fondazione IRCCS Ca’ Granda-Ospedale Maggiore Policlinico Hospital. The study was performed in compliance with the Declaration of Helsinki principles. Both patients were awake at the time of the decision and provided consent. An Istitutional Review Board protocol was approved by the Institutional Ethical Committee (Fondazione Genomic Study (FoGS), ethical approval 342-2020) and patients’ consent was obtained.

### 2.2. Patients’ Genotyping

Genotyping of the rs11385942(A) single nucleotide polymorphism, the top variant at the chromosome 3 cluster associated with the risk of severe COVID-19 [17], was performed in duplicate by custom-designed TaqMan 5′-nuclease assays (sequences available upon request).

### 2.3. Histopathologic Characterization

Routine-based histopathologic evaluation was performed on formalin-fixed and paraffin-embedded (FFPE) material from surgical specimens. Immunohistochemical staining was performed using an automated Dako OMNIS system (Dako-Agilent, Santa Clara, CA, USA). The ACE2 polyclonal antibody was from BIOSS Antibodies (code bs-1004R). Positive (small intestine) and negative (only secondary antibody) were included in the test (Appendix A). Quantification of lymphocyte subpopulations was performed using the Aperio digital slide scanner and an in-house developed algorithm within ImageScope (Leica Microsystems, Nussloch, DE, Germany) as previously described [18].

### 2.4. RNA Extraction from Lung Tissues, BAL, and PBMCs

Fresh lung samples from the two COVID-19 patients were manually dissected and homogenized before RNA purification. Briefly, total RNA from all fresh samples was isolated using the acid guanidium thiocyanate–phenol–chloroform method (RNAbee, Duotech, Milan, Italy) [19]. Bronchoalveolar lavage (BAL) sample was obtained from patient 1 at surgery and was compared to a non-COVID-19 lung-transplant subject affected by primary pulmonary hypertension. Total RNA from representative FFPE blocks of lung and hilar lymph nodes was extracted automatically using the RSC RNA FFPE kit by a Promega Maxwell instrument (Promega, Madison, WI, USA) and quantified with the Quantus fluorometer (Promega). Viral RNA was purified from the BAL sample and 8 × 10^5^ PBMCs using the Maxwell RSC Instrument with the Maxwell RSC Viral Total Nucleic Acid Purification Kit (both from Promega, Fitchburg, WI, USA) and eluted in RNAse-free water.

### 2.5. SARS-CoV-2 Quantification in Lungs, BAL, and Plasma Samples

RNA from fresh samples was reverse-transcribed using a single-step RT-qPCR (GoTaq 1-Step RT-qPCR; Promega, Fitchburg, WI, USA) on a CFX96 instrument (Bio-Rad, Hercules, CA, USA) using TaqMan probes specifically designed to target two regions of the nucleocapsid (N) gene of SARS-CoV-2 (2019-nCoV CDC qPCR Probe Assay emergency kit; IDT, Coralville, IA, USA), together with primers and probes for the human RNase P gene.

On FFPE samples, SARS-CoV-2 detection was performed using the CE-IVD Logix Smart COVID-19 kit (Co-Diagnostics, Salt Lake City, UT, USA), as previously described [20]. To avoid false-negative results, each sample was retested using SARS-CoV-2 RNA digital droplet PCR (ddPCR; QX200, Bio-Rad, Hercules, CA, USA) with CDC-approved primers and probes (Integrated DNA Technologies, Coralville, IA, USA), as previously described [21].

### 2.6. Isolation and Stimulation of PBMCs with SARS-CoV-2-Specific Antigens

Whole blood was collected from both patients in EDTA tubes (BD Vacutainer, San Diego, CA). Samples from patient 1 were obtained at three different points (T0 = pre-transplantation, T1 = 1 day post-transplantation, and T7 = 7 days post-transplantation). PBMCs were isolated by density gradient centrifugation on Ficoll (Cedarlane Laboratories Limited, Hornby, ON, Canada) and viable cells were counted with the automated cell counter ADAM-MC (Digital Bio, NanoEnTek Inc., Seoul, KR, Korea). PBMCs were resuspended at the concentration of 1 × 10^6^/mL in RPMI 1640 medium (Euroclone, Milan, Italy) supplemented with 10% fetal bovine serum, 1% of L-glutamine (LG), and 2% pen-streptomycin. Subsequently, PBMCs were stimulated with 500 ng/mL nucleocapsid- (N) and spike- (S) specific SARS-CoV-2 antigens (Novatein Biosciences, Cambridge, MA, USA). For gene expression and cytokine analyses, cells were harvested 10 h after stimulation and cytokine content was quantified on cell culture supernatants.

### 2.7. Quantigene Plex Gene Expression Assay

Gene expression was analyzed using the quantiGene Plex assay (Thermo Scientific, Waltham, MA, USA), which provides a fast and high-throughput solution for multiplexed gene expression quantitation, allowing the simultaneous measurement of 70 custom selected genes of interest in a single well of a 96-well plate.

### 2.8. Multiplex Cytokine Analysis

A 27-cytokine multiplex assay was performed on patients’ plasma and PBMC supernatants using magnetic bead immunoassays (Bio-Rad, CA, USA) and Luminex 100 technology (Luminex, Dallas, TX, USA) according to the manufacturer’s protocol. The same analysis was performed on the BAL fluid from patient 1 and a non-COVID-19 LTx subject (used as the control). We did not collect BAL and post-LTx plasma samples from patient 2 because of the presence of an infection from Klebsiella pneumoniae carbapenemase (KPC)-producing and uncontrolled sepsis.

## 3. Results

### 3.1. Clinical History

Patient 1 was an 18-year-old male with blood type A+, while patient 2 was a 48-year-old male with blood group B+. Both patients, who were previously healthy and without risk factors, fell ill with COVID-19 at the beginning of the pandemic in Italy. Conventional treatment (mechanical ventilation, prone positioning) failed after 16 days due to severe right pneumothorax with pneumatocele and to bilateral pneumothorax (patient 1). Both patients required mechanical support with veno-venous extracorporeal membrane oxygenation (ECMO; Avalon 31Fr, right internal jugular vein). The clinical condition of patient 1 became critical several times because of the high intrathoracic pressure that prevented flow through the extracorporeal system, severe pulmonary hypertension, systolic biventricular dysfunction, and unexpected thrombosis of the extracorporeal circuit despite systemic anticoagulation with direct thrombin inhibitors being performed. ECMO lasted for 55 days. Patient 2 had a stable course on ECMO (54 days); he developed severe pulmonary hypertension and right ventricular failure requiring specific treatment, such as with phosphodiesterase type 5 inhibitors, inotropes, calcium sensitizer, and inodilator. 

No recovery of the lung function was observed in both cases; therefore, the patients were listed for bLTx. The patients were bridged to lung transplantation on ECMO support (flow between 3–4 L/min, swipe gas 8 to 10 L/min) and assisted mechanical ventilation. Both patients also had pulmonary superinfections with Gram-positive and -negative bacteria, while SARS-Cov-2 RNA in nasopharyngeal swabs and BAL fluids became negative 30 days before listing.

### 3.2. Patients’ Genotype

The two patients did not carry the chromosome 3 rs11385942(A) risk variant previously associated with severe COVID-19. Conversely, both patients carried non-O blood groups, a characteristic related with increased risk of severe COVID-19 [17,22,23].

### 3.3. SARS-CoV-2 Presence in Lungs, Lymph Nodes, BAL, and Plasma Samples

In the first patient, SARS-CoV-2 RNA was identified in the lymph node but not in the lung tissue (Appendix A), while patient 2 tested negative on both specimens. ddPCR analysis confirmed the presence of SARS-CoV-2 RNA in the lymph node of patient 1 (54 copies/mL). Intriguingly, by ddPCR (Appendix A), low copies of viral RNA (region N1) were detected also in the lung parenchyma of patient 1 (12 copies/mL). BAL and plasma samples from both patients were negative for SARS-CoV-2 RNA.

### 3.4. Imaging and Pathologic Findings 

At radiological examination, both patients showed a bilateral parenchymal involvement, with diffuse fibrosis and pneumothorax (Figure 1A,B). Furthermore, a voluminous pneumatocele was also present in the left upper lobe of patient 1. At gross examination, lungs from patient 1 displayed features of so-called hepatization, with foci of hemorrhage and consolidation (Figure 1C). Lungs from patient 2 showed a bronchiectasis picture with parenchymal thickening, with small peripheral foci of hemorrhage and emphysema (Figure 1D). For comparison, a lung without end-stage chronic disease is presented in Appendix A.

At the microscopic level (Figure 1E–J), both cases displayed widespread features of proliferative DAD with prominent fibroblastic reaction and almost complete type 2 pneumocyte hyperplasia and frequent cytopathic changes (Figure 1G–J), with only focal presence of microthrombi (Figure 1H). Fibrotic-phase DAD was particularly observed in patient 2, featuring a honeycomb-like pattern (Figure 1F,I,J) reminiscent of a fully established pattern of usual interstitial pneumonia (UIP). Foci of peripheral emphysema and hemorrhage were present, although the latter was more extensive for patient 1, who also displayed features of superimposed pneumonia. Moderate inflammatory infiltrate was present in both cases (Figure 2A), mostly in the form of perivascular T lymphocytes (Figure 2B), with a prevalence of the CD8+ T cells with respect to the CD4+ T-cell component (Figure 2C–E) and rare CD56+ natural killer cells. In regards to the B-cell infiltrate, scattered CD20+ lymphocytes and MUM1+ plasma cells were observed (Figure 2A,F). Examination of hilar lymph nodes (Appendix A) revealed in both cases a slightly depleted B-cell compartment (Appendix A), at times with regressed follicular germinal centers and an expanded paracortical zone with actively proliferating (Ki67+) MUM1+ immunoblasts (Appendix A) and plasma cells skewed towards the IgG class phenotype. This immunophenotype was accompanied by a well-represented T-cell compartment with both CD3+ T-cell lineages (Appendix A). Finally, lymph nodes showed a venular component with high endothelia, with a focal angioendotheliomatosis-like pattern (Appendix A).

### 3.5. Immune Gene Profile of Lung Tissues 

Globally, lungs from COVID-19 patients displayed an immune-activated profile (Figure 3A). Such hyperactivation was far more evident in the lungs from patient 1, with sharp upregulation of the genes involved in different aspects of the antiviral immune response (Figure 3B). This signature included effector cytokines and chemokines; pathogen recognition receptors, in particular the Toll-like receptors TLR3 and TLR7 involved in virus detection; cholesterol metabolism; immune activation receptors; and inflammasome components (Figure 3B and Appendix A). Furthermore, patient 1 featured higher expression of the receptors involved in SARS-CoV-2 infection, ACE2, and the virus-interacting protein AGTR1 compared with patient 2 (Figure 3B). ACE2 expression in the lung parenchyma of COVID-19 LTx patients was also evaluated at the protein level. Indeed, by immunohistochemistry, we found that ACE2 was present both in the alveolar space and the primary bronchus, besides being expressed by small vessels (internal positive control; Figure 3C).

### 3.6. Cytokine and Chemokine Levels in Plasma and BAL Samples 

In line with the gene expression pattern observed in the lungs, plasmatic proinflammatory antiviral cytokines and chemokines were upregulated in patient 1 compared with patient 2 (Figure 4A). Specifically, the interleukin-1β (IL-1β) and IL-6 were, respectively, 4.9 and 5.8 times higher in patient 1 compared to patient 2 (Figure 4A and Appendix A). Analyses performed in the BAL fluids showed that the chemokines CCL3 and CCL5 and the proinflammatory molecules IL-1β and IL-6 were significantly higher in patient 1 compared to the non-COVID-19 bLTx patient (Appendix A). Specifically, in patient 1, we observed a common signature in the plasma and in the BAL of upregulated signaling molecules involved in macrophage activation, which included chemoattractant cytokines (CCL3, CCL5, MCP-1), interleukins, and interferon-γ (IFN-γ; Figure 4B and Appendix A). Lastly, analysis of T1 and T7 plasma samples from patient 1 showed a quick and progressive decrease of chemokines and proinflammatory cytokines starting as early as one day post-bLTx (Figure 4C). 

### 3.7. Immune Profile of Stimulated PBMCs 

At the gene expression level, PBMCs from both COVID-19 LTx patients displayed a strong immune activation profile both at basal condition (Figure 5A) and upon stimulation with SARS-CoV-2-specific antigens (Figure 5B). Notably, such hyperactivation status was higher in patient 1, with upregulation of effector cytokines and chemokines, pathogen recognition receptors, immune activation receptors, and inflammasome components (Figure 5B and Appendix A). In keeping with the gene expression data, quantification of cytokines in PBMC supernatants confirmed that circulating lymphocytes from both patients were characterized by a proinflammatory phenotype (Figure 5C). In particular, high levels of the monocyte/macrophage chemotactic and activating factors (such as CCL3, CCL4, G-CSF and IFN-γ) and the proinflammatory cytokines IL-6 and TNF-α were detected in both individuals (Figure 4C and Appendix A).

Finally, stimulation of PBMCs from patient 1 at days 1 and 7 post-LTx resulted in a cytokine profile consistent with the maintenance of an antiviral-specific immune memory despite the immunosuppressive treatment (Figure 5D and Appendix A).

## 4. Discussion

In this report, we describe the immunopathological phenotype detected in the lungs and PBMCs from two adult males who underwent bLTx after end-stage COVID-19. Both patients carried non-O blood groups, a genotype associated with an about two-fold higher risk of developing respiratory failure and requiring mechanical ventilation during COVID-19 [17]. However, they were previously healthy individuals, and both were negative for the rs11385942(A) allele at the chromosome 3 gene cluster, which is the main commonly inherited determinant of the risk of severe COVID-19 [17].

To date, only a few reports of bLTx in COVID-19 patients have been described [24,25,26], with the notable case of a young patient who underwent transplantation despite the persistence of SARS-CoV-2 RNA positivity at nasal swabs [25]. Similarly to previous reports, the decision to proceed to transplantation in both of our cases was based on the evaluation of the unlikely recovery of the lung function after a protracted course of COVID-19 in patients with single organ (pulmonary) failure.

Our data show that the removed lungs displayed, at histopathologic level, extensive fibrotic remodeling of the respiratory tract in addition to focal features secondary to complications of the disease, probably due to the superimposed infections. Overall, these features mimic what is observed for usual interstitial pneumonia, but in the COVID-19 cases, this involves the whole lungs with almost no sparing of tissue. In contrast with most of the previous literature [5,6,7,8,10], we could not document the elementary lesions of acute DAD, and thrombotic microangiopathy was marginal. This could be related to the fact that previously published series were collected during autopsies following acute disease, whereas we analyzed lungs from live patients after end-stage COVID-19. On the contrary, our data confirm the observations of Bharat and coworkers [26], as the clinico–pathologic profile at the terminal stages in our patients resembled the picture of UIP. It should be stressed that our findings highlight the potential output of a longstanding course of severe disease as a condition with very limited potential for tissue recovery, rather than providing clues on the physiopathology of the SARS-CoV-2-mediated damage on respiratory tissues.

The role of inflammation in COVID-19 physiopathology and severity is still under scrutiny [1,15]. In our patients, we observed lung parenchyma with predominant CD8+ T-lymphocyte infiltration, compatible with the late-fibrosing stages of DAD and lung interstitial remodeling. In line with this, no or very low (in patient 1 by ddPCR) viral load could be documented in the lungs. Examination of the draining (hilar) lymph nodes showed the expansion of the immunoblasts, a pattern usually observed in lymphadenitis following an acute (viral) stimulus. Further, we observed features of the reticular endothelial system activation in the form of small vessel proliferation. This pattern of lymphoid hyperplasia stands in line with the higher levels of IL-6 observed in BAL and plasma from patient 1, as the molecule exerts a proinflammatory function favoring the activation of antigen-selected B cells and antibody production [27].

At the molecular level, the gene expression profile of lungs and PBMCs samples showed hyperactivation of inflammatory pathways, including the pattern recognition receptors involved in the antiviral response, chemokines and cytokines such as TLR3, TLR7, CCL5, CXCL10, IL-1β, IL-6, IL-7, IL-22 and TNF-α. This molecular setting was more pronounced in patient 1 compared with patient 2. In keeping with this, the BAL fluids from patient 1 had higher levels of CCL3, CCL5, IL-1β, IL-6, MCP-1 and IFN-γ, while his PBMCs maintained an antiviral-specific immune memory even after LTx immunosuppressive therapy. Though suggestive, these findings mirror what we observed at the histopathologic level, that is, a more pronounced inflammatory pattern in patient 2. Indeed, we documented an incipient end-stage DAD for patient 1, as compared with the overtly fibrotic pattern with an almost spent cellular response for patient 2. Collectively, these data may suggest that a more activated inflammatory profile, which can be easily assessed on PBMCs, may be supportive of a less advanced stage of lung tissue remodeling, although the limited number of cases tested should prompt further investigation.

The finding of residual SARS-CoV-2 RNA copies in the removed tissues from patient 1, despite the negativity of nasal swabs and BAL at the time of surgery, might explain the hyperactivated inflammatory state detected in this subject. However, it should be stressed that the RNA presence does not equate to vital viral copies. Collectively, our findings support, both at the molecular and histopathologic levels, the key role of a state of hypercytokinemia (the so-called “cytokine storm”) in the pathogenesis of COVID-19, which may translate in the potential benefit of immune-modulating agents in preventing tissue damage [28].

In conclusion, although the specific factors promoting progression to irreversible lung damage in COVID-19 patients remain undetermined, we provide new insights into the pathologic evolution of end-stage SARS-CoV-2-affected lungs. We show that COVID-19 potentially outputs in a new, UIP-like condition of end-stage pulmonary disease, which may affect a relevant proportion of patients who experienced severe SARS-CoV-2 infection [11]. Therefore, our study, although preliminary, highlights that a COVID-19-related UIP-like condition may represent a new pathological entity eventually suitable for a life-saving procedure such as lung transplantation.

## Figures and Tables

**Figure 1 cells-10-00611-f001:**
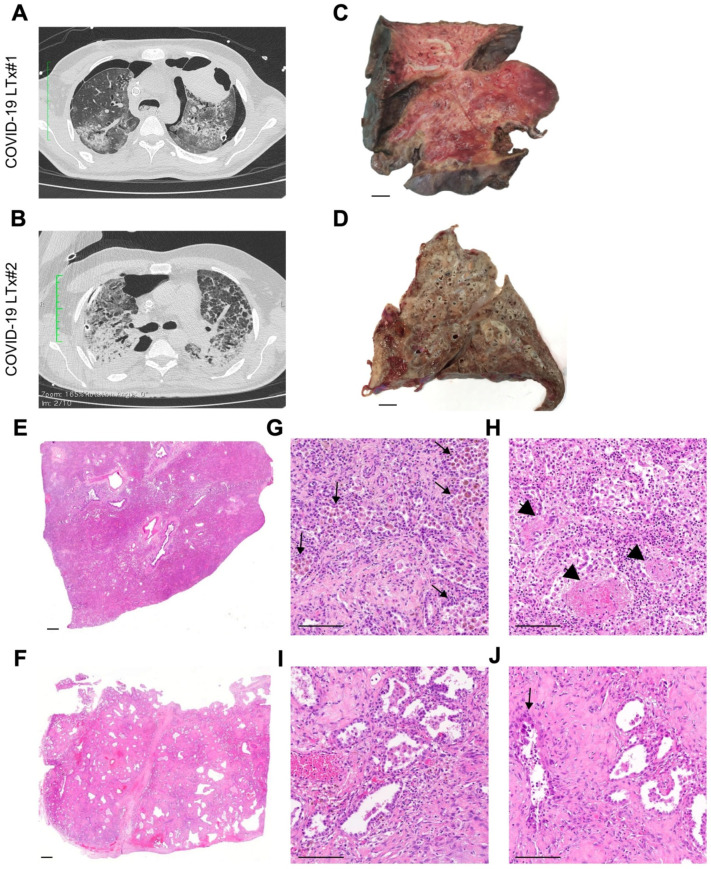
The histo-radiological phenotype of end-stage COVID-19 lungs. (**A**,**B**) Computed tomography images of both patients show a bilateral parenchymal involvement, with diffuse fibrosis and pneumothorax; in the first case, a voluminous pneumatocele is also present in the left upper lobe. (**C**,**D**) Macroscopic examination of the removed lungs shows widespread derangement of the parenchyma with signs of hepatization (especially in patient 1; **C**) or fibrosis (especially in patient 2; **D**). Microscopic analysis evidences features of the proliferative/organizing phase of diffuse alveolar damage alternated with foci of overtly fibrotic/late-stage interstitial pneumonitis (**E**,**F**), with inflammatory cells such as histiocytes and granulocytes filling the alveolar spaces (**G**, arrows), and scattered presence of fibrinoid necrosis foci (**H**, arrowheads). Lungs from patient 2 displayed a diffuse process as well, with initial features of honeycombing (**F**,**I**), reminiscent of an interstitial pneumonia-like process. Type 2 pneumocyte hyperplasia with epithelial atypia (**J**, arrow) was also detected. Scale bars: 1 cm (**C**,**D**); 1 mm (**E**,**F**); 100 μm (**G**–**J**).

**Figure 2 cells-10-00611-f002:**
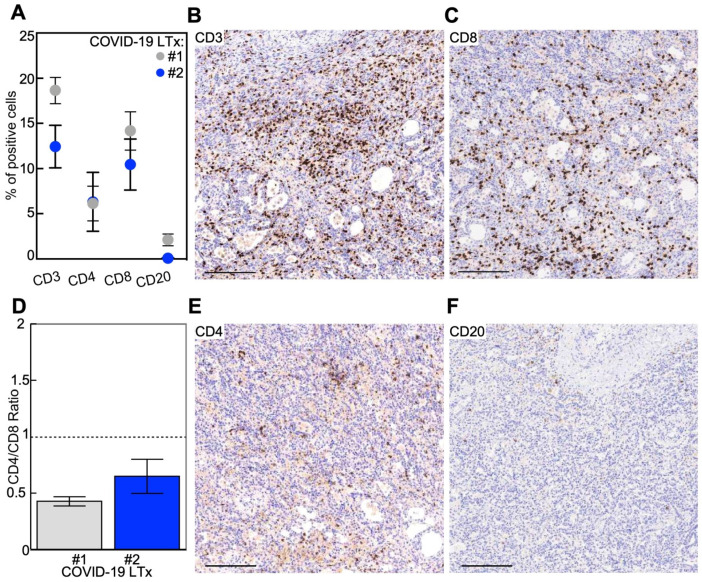
The immunopathological phenotype of end-stage COVID-19 lungs. Immunophenotype of the removed lungs from COVID-19 lung transplantation (LTx) shows the predominant presence of CD8+ T cells (**A**–**F**). The indicated lymphocytic subsets were quantified using the Aperio algorithm in three representative lung areas of the two patients (**A**). Dots: mean ± SEM. Representative images of CD3- (**B**), CD4- (**E**), CD8- (**C**) or CD20-positive (**F**) infiltrates are shown. The CD4/CD8 ratio was calculated for both patients (**D**; bars, mean ± SEM). The dotted line indicates the physiological value. Scale bar: 100 μm.

**Figure 3 cells-10-00611-f003:**
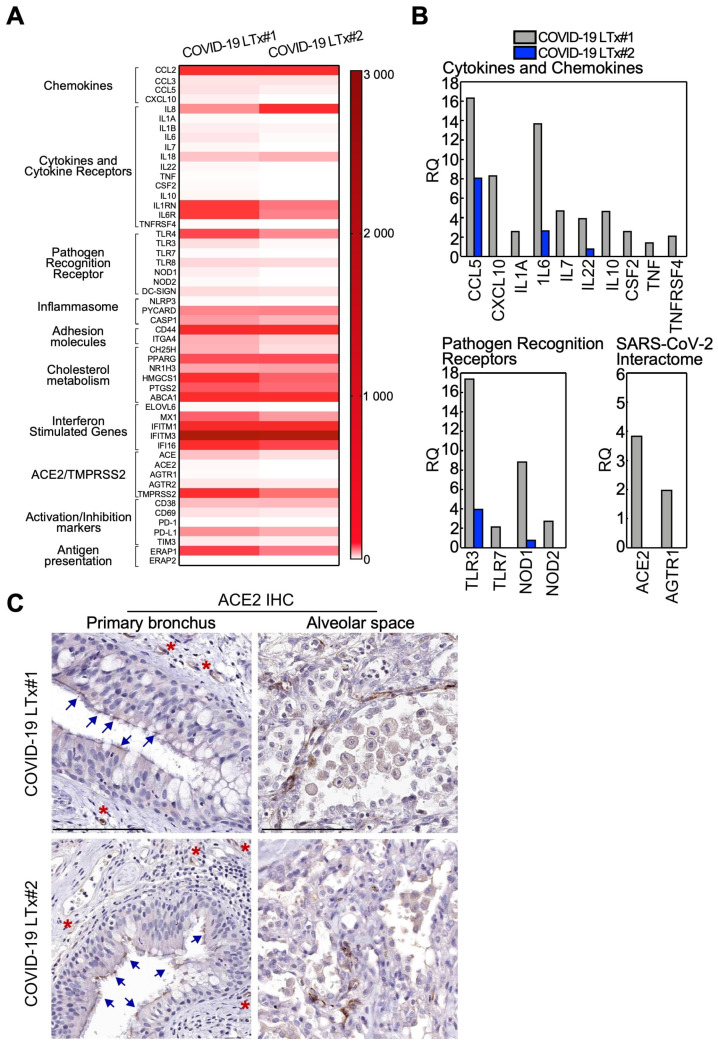
Immune gene profiles of end-stage COVID-19 lung tissues. (**A**) Heatmap of all analyzed genes in the lungs of LTx patients. (**B**) Genes upregulated (>2-fold) in LTx patient 1 compared with LTx patient 2 are shown (see also Appendix A for details). RQ: relative quantity. (**C**) ACE2 expression in the removed lungs was analyzed by IHC and revealed by DAB (brown color). Arrows indicate ACE2-positive ciliated cells in the primary bronchus, while asterisks indicate stained vessels (positive internal control). Scale bar: 100 μm.

**Figure 4 cells-10-00611-f004:**
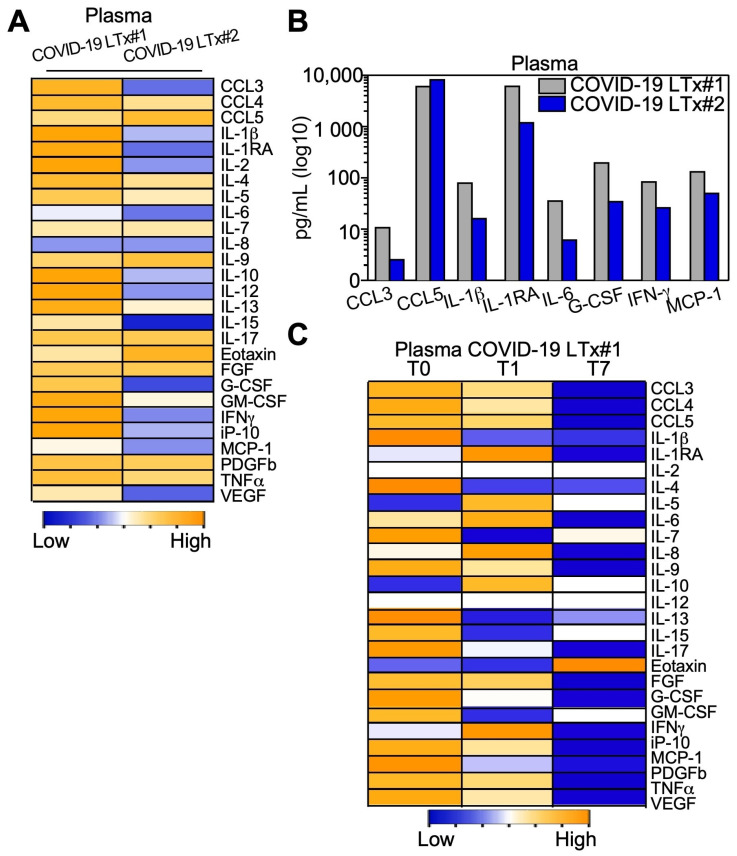
Circulating cytokine and chemokine profiles of end-stage COVID-19 lung tissues. The indicated molecules were quantified in the plasma of COVID-19 LTx patients (**A**–**C**). (**B**) Cytokines and chemokines over-represented in the plasma of COVID-19 LTx patient 1 compared with LTx patient 2 are shown. (**C**) The indicated cytokines and chemokines were quantified in plasma samples from COVID-19 LTx patient 1 at 1 (T1) or 7 (T7) days after transplantation (T0, at surgery). Yellow and blue color in the heatmaps show a high or low level, respectively (see also Appendix A for details).

**Figure 5 cells-10-00611-f005:**
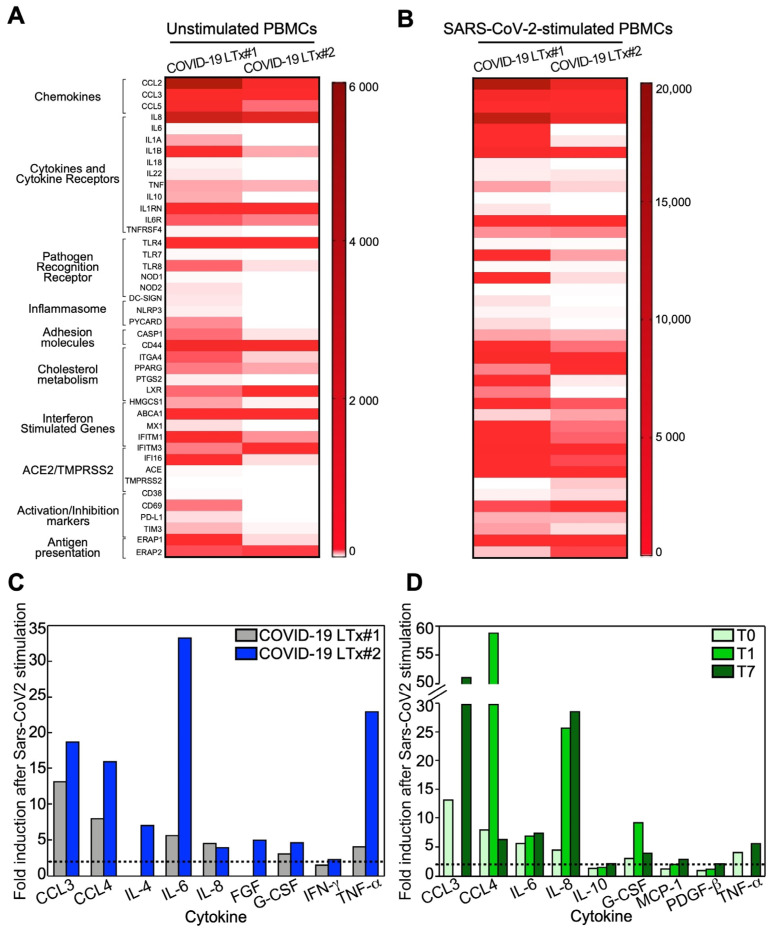
Immune gene signatures of SARS-CoV-2-stimulated PBMCs. Immune gene status was analyzed in PBMCs from the two COVID-19 LTx patients at baseline (**A**) or after stimulation with SARS-CoV-2 nucleocapsid and spike antigens (**B**; see also Appendix A for details). (**C**,**D**) Cytokine and chemokine secretion was evaluated in supernatants of PBMCs collected from both patients at surgery (**C**) or in LTx patient 1 at 1 (T1) or 7 (T7) days after transplantation (T0; see also Appendix A for details).

## Data Availability

The data presented in this study are available within the manuscript or in the Appendix A available online.

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
