# Peer review of "Emergency Lung Transplantation after COVID-19: Immunopathological Insights on Two Affected Patients"

_cells, 2021, doi:10.3390/cells10030611_

Round 1

Reviewer 1 Report

In this article by Croci and colleagues, the authors provide evidence for the underlying mechanisms of COVID-19 related lung failure in 2 patients who underwent lung transplantation as their final treatment option. The authors documented signs of irreversible fibrotic changes on the explanted lungs and hyperinflammatory state on both patients elegantly. The authors should be congratulated for their contribution to our understanding about COVID-19 related lung failure. I have the following suggestions for the clarity and improvement.

  1. How many patients were evaluated for lung transplantation related to COVID-19 induced fibrosis during the study period? When did you exactly listed the patients and what was the waiting time while on the list? It would be helpful to provide a timeline indicating the course of disease before and after bLTx for both patients including postoperative course as a supplementary figure.
  2. How did you obtain consent in such critically ill patients? Were they awake at the time of evaluation? Performing a transplant without first-person consent carry important risks for the care after bLTx. Please provide a detailed information on the article.
  3. What about the post-operative course? It appears that the patient #1 was alive on day 7 from a blood sample. Are these recipients still alive? How many days did they require ECMO-support post-transplant, duration of ICU and hospitalization? It is important to give information about postoperative course when the knowledge is scarce in these unique LTx recipients.
  4. The limitation of this case report should be stated. The knowledge acquired from this article is belonging to 2 patients only. Moreover, there are missing samples belonging to the patient #2. The reasons of this should be stated in the text.
  5. There are marked differences between patient 1 and 2 when we looked at the macroscopical and microscopical findings as well as the inflammatory profiling. Moreover, response to stimulation on PBMCs was much higher in patient 2 than in patient 1? Although patient 2 had a markedly less inflammatory response at baseline. Can you elaborate on that? How do the authors explain this discrepancy? Could there be different phenotypes of COVID related lung fibrosis?
  6. Overall presentation of plasma sample results at T0 and baseline on Fig 3 is confusion. Are they belonging to the same time point? When is exactly your T0? Is it at surgery before the incision? If the T0 and baseline samples are the same, why the heatmap is different between 3A and 3E for patient 1? Similarly, T0 and baseline time points are confusing for fig 4 and supplementary table 4. Please clarify on the legends.
  7. Also, on figure 3, the results of a BAL sample from a Non-COVID recipient should not be presented. It is confusing. You may present in the supplementary data only. Please indicate the BAL samples from patient 2 is missing and present the results of Patient 1 only.
  8. It is not possible to observe the microscopical findings on Fig 1 due to the small size of the images. Fig 1 can be divided into two parts, first 2 column as one figure and the second 2 column another one. Or creating a link that opens up a better resolution of images would be helpful. Furthermore, figures should be printed on a single page with its legend whenever possible.
  9. Please illustrate findings on the sections of supplementary fig 3 as well.
  10. Minor:
  • Page 2 line 47: I would use at the “microscopical” level
  • Discussion line 289: I would not use “young” for a 48yo recipient, rather I would say “two adult males”
  • Please change to 3rd column of table S2 as non-COVID PLASMA, and the 1st column COVID-PLASMA
  • Typo: Supplementery Fig 2 legend: lung samples letter should be (D)

Author Response

Ms. cells-1130567: “Emergency lung transplantation after COVID-19: immunopathological insights on two affected patients”

We thank the Editor and the Reviewer for considering our manuscript and finding it potentially interesting for Cells. Here are our point-by-point responses (in blue) to the Reviewers comments. Modifications in the text are tracked for presentation clarity.

Reviewer 1

  1. How many patients were evaluated for lung transplantation related to COVID-19 induced fibrosis during the study period? When did you exactly listed the patients and what was the waiting time while on the list? It would be helpful to provide a timeline indicating the course of disease before and after bLTx for both patients including postoperative course as a supplementary figure. We acknowledge and appreciate the scientific curiosity of the Reviewer. Indeed, at the end of the first pandemic, we evaluated for bilateral lung transplantation six end-stage COVID-19 patients but only the two males reported here were finally listed respectively on April 4 (LT#1) and June 4 2020 (LT#2). The remaining four were discharged because of multi-organ dysfunction, age >50y and uncontrolled sepsis. LT#1 patient stayed 18 days on the waiting list, while LT#2 stayed 5 days. Clinical details about the pre- and post-operative course of these two patients are specified in a separate publication in Lancet Respiratory Medicine, which will soon be published soon in press (the title of the manuscript is: “Lung transplantation for COVID-19 associated Acute Respiratory Distress Syndrome – International experience of the first national transplants”).

  1. How did you obtain consent in such critically ill patients? Were they awake at the time of evaluation? Performing a transplant without first-person consent carry important risks for the care after bLTx. Please provide a detailed information on the article. Both patients were awake at the time of the bLT decision and were informed about the medical procedure. This is detailed together with the IRB approval in the Methods, Patients section at page 12.
  2. What about the post-operative course? It appears that the patient #1 was alive on day 7 from a blood sample. Are these recipients still alive? How many days did they require ECMO-support post-transplant, duration of ICU and hospitalization? It is important to give information about postoperative course when the knowledge is scarce in these unique LTx recipients. We acknowledge that the post-operative course is important to implement the scientific understanding of the clinical outcome in end-stage COVID19 patients treated with LT. As stated before, these data are part of a separate article that will be appear in Lancet Respiratory Medicine.
  3. The limitation of this case report should be stated. The knowledge acquired from this article is belonging to 2 patients only. Moreover, there are missing samples belonging to the patient #2. The reasons of this should be stated in the text. We concur with the Reviewer and we acknowledged these limitations at page 11. Unfortunately, post-LT fluids from patient 2 could not be sampled because of the presence of superimposed respiratory infection by KPC (resistant Klebsiella Pneumoniae) and uncontrolled sepsis. This is specified at page 3 in the Methods.
  4. There are marked differences between patient 1 and 2 when we looked at the macroscopical and microscopical findings as well as the inflammatory profiling. Moreover, response to stimulation on PBMCs was much higher in patient 2 than in patient 1? Although patient 2 had a markedly less inflammatory response at baseline. Can you elaborate on that? How do the authors explain this discrepancy? Could there be different phenotypes of COVID related lung fibrosis? We thank the Reviewer for making this point. The different pattern of tissue damage described in the two patients is certainly a major point of interest. Major limitations to any elaboration on that include the low number of cases and the fact that we focused on assessing the pattern of inflammation and cellular response to the viral load, rather than on the signatures related to macrophage activation and fibrosis. However, as highlighted in the results and commented in the discussion of the revised version, the pattern of tissue damage in patient 1 featured a more pronounced cellular response, with only incipient end stage DAD, corresponding the hyperactivated profile. This is also supported by the gene signatures obtained from the lung and PBMCs of patient#1. On the contrary, patient 2 featured an almost spent phase of inflammation, with a well-established fibrosis.
  5. Overall presentation of plasma sample results at T0 and baseline on Fig 3 is confusion. Are they belonging to the same time point? When is exactly your T0? Is it at surgery before the incision? If the T0 and baseline samples are the same, why the heatmap is different between 3A and 3E for patient 1? Similarly, T0 and baseline time points are confusing for fig 4 and supplementary table 4. Please clarify on the legends. We concur with the Reviewer that this point and figure was unclear. Accordingly, we revised the panels and the figure legend (new Figure 4), and the time of the T0 sample (at surgery) is specified. Nevertheless, since we used a heatmap to show the data, a variation in the colours’ shades is dependent on the samples included in each analysis, so the heatmap in panel A might not overlap completely to the one in panel C because of the different samples shown.

  1. Also, on figure 3, the results of a BAL sample from a Non-COVID recipient should not be presented. It is confusing. You may present in the supplementary data only. Please indicate the BAL samples from patient 2 is missing and present the results of Patient 1 only. According to the Reviewer suggestion we moved this part of the figure to the Supplementary data (new Supplementary Figure 6). The reason why BAL from pt#2 is missing is the presence of uncontrolled sepsis. This is now indicated in the Methods section, page 3.

  1. It is not possible to observe the microscopical findings on Fig 1 due to the small size of the images. Fig 1 can be divided into two parts, first 2 column as one figure and the second 2 column another one. Or creating a link that opens up a better resolution of images would be helpful. Furthermore, figures should be printed on a single page with its legend whenever possible. We concur with the Reviewer that panels were difficult to see. Accordingly, Figure 1 has been split into two separate figures and histological images have been enlarged.

  1. Please illustrate findings on the sections of supplementary fig 3 as well. We thank the Reviewer for this suggestion and we amended the text (Page 4 of the Results).

All minor comments and typos have been addressed

Reviewer 2 Report

The authors present a timely clinical report on the cellular state of human lungs from two patients that had active Covid infection. This study focuses on the immune infiltration of the human lung after infection and gives a sense of the cellular changes in the pathology of the Covid-19 disease progression.

Comments:

  • While this paper is targeted to an audience of lung biologists I believe text editing should focus on comparison to the normal lung, if possible comparable image from normal lung should be shown (Figure 1). Given the broad audience of cell biologists that read this journal that would be highly impactful
  • In figure 2, the ACE2 antibody staining needs negative control data – IgG staining should suffice in the neighboring lung sections.

Author Response

Ms. cells-1130567: “Emergency lung transplantation after COVID-19: immunopathological insights on two affected patients”

We thank the Editor and the Reviewer for considering our manuscript and finding it potentially interesting for Cells. Here are our point-by-point responses (in blue) to the Reviewers comments. Modifications in the text are tracked for presentation clarity.

Reviewer 2

  1. While this paper is targeted to an audience of lung biologists I believe text editing should focus on comparison to the normal lung, if possible comparable image from normal lung should be shown (Figure 1). Given the broad audience of cell biologists that read this journal that would be highly impactful. We thank the Reviewer for making this point. A representative image from a lung without chronic end-stage respiratory disease in now presented as supplementary material (Supplementary Figure 4).
  2. In figure 2, the ACE2 antibody staining needs negative control data – IgG staining should suffice in the neighboring lung sections. We thank the Reviewer for making this point. Accordingly, positive and negative controls have been included as Supplentary Material (new Supplementary Figure 1A and B, respectively).
